# Investigating the vertical extent of the 2023 summer Canadian wildfire impacts with satellite observations

Selena Zhang[1], Susan Solomon[1], Chris D. Boone[2], Ghassan Taha[3,4]

[1]Department of Earth, Atmospheric, and Planetary Sciences, Massachusetts Institute of Technology, Cambridge, MA 02139, USA
[2]Department of Chemistry, University of Waterloo, Waterloo, Ontario N2L 3G1, Canada
[3]Morgan State University, Baltimore, MD 21251, USA
[4]NASA Goddard Space Flight Center, Greenbelt, MD 20771, USA

*Correspondence to*: Selena Zhang (selenaxz@mit.edu)

**Abstract.** Pyrocumulonimbus clouds (pyroCbs) generated by intense wildfires can serve as a direct pathway for the injection of aerosols and gaseous pollutants into the lower stratosphere, resulting in significant chemical, radiative, and dynamical changes. Canada experienced an extremely severe wildfire season in 2023, with a total area burned that substantially exceeded those of previous events known to have impacted the stratosphere (such as the 2020 Australian fires). This season also had record-high pyroCb activity, which raises the question of whether the 2023 Canadian event resulted in significant stratospheric perturbations. Here, we investigate this anomalous wildfire season using retrievals from multiple satellite instruments, ACE-FTS (Atmospheric Chemistry Experiment – Fourier Transform Spectrometer), OMPS LP (Ozone Mapping and Profile Suite Limb Profiler), and MLS (Microwave Limb Sounder) to determine the vertical extents of the wildfire smoke along with chemical signatures of biomass burning. These data show that smoke primarily reached the upper troposphere and only a nominal amount managed to penetrate the tropopause. Only a few ACE-FTS occultations captured elevated abundances of biomass burning products in the lowermost stratosphere. OMPS LP aerosol measurements also indicate that any smoke that made it past the tropopause did not last long enough or reach high enough to significantly perturb stratospheric composition. While this work focuses on Canadian wildfires given the extensive burned area, pyroCbs at other longitudes (e.g. Siberia) are also captured in the compositional analysis. These results highlight that despite the formation of many pyroCbs in major wildfires, those capable of penetrating the tropopause are extremely rare; this in turn means that even a massive area burned is not necessarily an indicator of stratospheric effects.

## 1 Introduction

In 2023, a record-breaking fire season burned over 15 million hectares in Canada, over seven times the 1983 – 2022 annual average burned area of 2.1 million hectares (Canadian Interagency Forest Fire Centre Inc., https://ciffc.net/statistics/). These events follow a trend of increasingly extensive and destructive wildfires, often referred to as megafires, that are projected to become more frequent under a changing climate (Di Virgilio et al., 2019; Williams et al., 2019; Pausas and Keeley, 2021). In

addition to the well-studied impacts of megafires on air quality and tropospheric composition, a number of events have also injected wildfire smoke into the stratosphere via deep convective events known as pyrocumulonimbus clouds (pyroCbs) (Fromm et al., 2010; Fromm et al., 2019; Fromm et al., 2022).


PyroCbs are towering thunderstorms triggered by intense surface fire activity that also require specific meteorological conditions for development (including very dry surface conditions and high moisture and instability in the mid-troposphere, see Peterson et al., 2017). Strong evidence has been presented to show that past wildfire events injected significant amounts of smoke above the tropopause and that smoke-charged vortices may also self-loft (Khaykin et al., 2020; Renard et al., 2020;

Lestrelin et al., 2021, Sellitto et al., 2023). Perhaps the most notable example of these effects is the 2019 – 2020 Australian New Year Super Outbreak (ANYSO) event that injected up to 1.1 Tg of smoke into the stratosphere (Peterson et al., 2021). ANYSO, which was part of the Australian "Black Summer" bushfire season where around 5.8 million hectares burned, has been linked to stratospheric ozone depletion and numerous climate impacts (Boer et al., 2020; Kablick et al., 2020; Rieger et al., 2021; Bernath et al., 2022; Solomon et al., 2023). While the Black Summer fires raged for about seven months, nearly all

the stratospheric input occurred on just a few days; December 29-31, 2019 and January 4, 2020. (Davey and Sarre, 2020; Khaykin et al., 2020; Peterson et al., 2021). Stratospheric perturbations also occurred as a result of the 2017 Pacific Northwest Event (PNE) in British Columbia, when on August 12, 2017, pyroCbs injected an estimated 0.3 Tg of aerosols into the stratosphere (Peterson et al., 2018; Torres et al., 2020). For context compared to 2023, the 2017 Canadian wildfires burned a total of 3.5 million hectares, with over 1.2 million hectares burned in British Columbia (Government of British

Columbia, https://www2.gov.bc.ca/).

Given the outsized area burned in the 2023 Canadian wildfires, significant stratospheric impacts perhaps as in the PNE or ANYSO events might be expected. It has been reported that at least 135 pyroCbs occurred in Canada between May and August 2023, a record-high amount further suggesting the possibility of substantial perturbations to stratospheric

composition (Smith, 2023). Recent research has investigated the tropospheric impacts of the 2023 fires, identifying record-high particulate matter emissions with implications for air quality and human health, but the vertical extent of these impacts and whether smoke entered the stratosphere has not yet been reported (Thurston et al., 2023; Wang et al., 2023).

The Atmospheric Chemistry Experiment – Fourier Transform Spectrometer (ACE-FTS) is a satellite instrument that detects

a number of species including multiple biomass burning tracers with high sensitivity and precision (Sect. 2.1). Here, we use data from ACE as well as aerosol extinction data from the Ozone Mapping and Profiler Suite Limb Profiler (OMPS LP), described in Sect. 2.2, to investigate whether the 2023 Canadian wildfires perturbed stratospheric composition. MLS carbon monoxide data is also analyzed.

## 2 Data and methods

### 2.1 ACE-FTS and MLS data

ACE-FTS aboard the Canadian ACE/SCISAT-1 platform is a solar occultation instrument that collects up to 30 daily atmospheric absorption measurements at sunrise and sunset using the Sun as a light source (Bernath, 2005; Bernath, 2017). The infrared Fourier transform spectrometer measures over a wide spectral range (750 to 4400 cm$^{-1}$) with a high resolution of 0.02 cm$^{-1}$ and signal-to-noise ratio ranging between 100:1 and 400:1 (Buijs et al., 2013). The ACE-FTS processing version 5.2 provides vertical volume mixing ratio (VMR) profiles for 46 molecules and 24 isotopologues including characteristic biomass burning indicator molecules such as carbon monoxide (CO), hydrogen cyanide (HCN), acetonitrile (CH$_3$CN), and ethane (C$_2$H$_6$) (Boone et al., 2023). A pair of filtered imagers also measures atmospheric extinction at two wavelengths: visible (VIS, 527.11 nm) and near-infrared (NIR, 1020.55 nm). The NIR imager is less likely to become saturated in cases of strong aerosol extinction and was thus used in this analysis for detection of aerosol loads (Vanhellemont et al., 2008; Boone et al., 2020). Temperature profiles are also collected with every occultation and were used to calculate tropopause heights (Sect. 2.3). Data from ACE are available at https://databace.scisat.ca/level2/ace_v5.2/ starting from 2004.

Given the limited data coverage of ACE, we also analyze carbon monoxide data from the Microwave Limb Sounder (MLS) aboard the NASA Aura satellite (Waters et al., 2006) given its higher spatial coverage. The MLS instrument has seven radiometers that measure microwave thermal emissions between 118 GHz and 2.5 THz from the limb of the atmosphere to determine vertical profiles of atmospheric constituents. This instrument has a much higher spatial coverage than ACE, but the use of microwave spectroscopy provides data with a lower signal-to-noise ratio on individual soundings and a more limited vertical range. For example, the CO data product is recommended for scientific use between 215 – 0.001 hPa, while the HCN and CH$_3$CN valid ranges are even smaller at 21 – 0.1 hPa and 46 – 1.0 hPa respectively (Livesey et al., 2022). Since we are interested in the vertical profiles of biomass burning products, MLS is not as useful as ACE for our analysis given the lack of reliable data at our pressure range of interest in the lowermost stratosphere and its transition to the tropopause and upper troposphere. However, carbon monoxide data from MLS in the lower stratosphere is still useful as complementary data since strong signals from large perturbations would be clearly detected (Fig. S1). Additionally, comparison of ACE and MLS profiles in our latitude range of interest supports our conclusions from ACE data despite more limited coverage (Fig. S2). Data from MLS are available at https://disc.gsfc.nasa.gov/datasets/ML2CO_NRT_005/summary starting from 2004.

### 2.2 OMPS LP aerosol data

OMPS LP on the Suomi NPP satellite is a limb profiler that measures scattered UV, visible, and near-IR radiation. Aerosol extinction coefficients are retrieved for six wavelengths (510, 600, 675, 745, 849, and 997 nm) with the V2.1 algorithm (Taha et al., 2021). OMPS LP measures along Earth's limb with three parallel vertical slits, one central slit that views along

the nadir track and two side slits viewing with a cross-track separation of 250 km at the tangent point allowing for near-global daily coverage. The 745 nm channel was used in this analysis given its high sensitivity to aerosol loading and low bias, and aerosol extinction is analyzed as an indicator for aerosol abundance. A cloud detection algorithm detects the highest cloud altitude and flags all aerosol extinction measurements below the peak cloud level, allowing for a cloud-filtered data product (Chen, 2016). The retrieved aerosol-to-molecular extinction ratio analyzed in this work is analogous to an aerosol mixing ratio, see Loughman et al., 2018 and Taha et al., 2022 for a detailed account of the retrieval algorithm and previous data applications. Data from OMPS is available at https://disc.gsfc.nasa.gov/datasets/OMPS_NPP_LP_L2_ AER_ DAILY_2/summary.

## 2.3 Tropopause height calculation

Determination of stratospheric entry depends on the definition of tropopause height. Given the high variability of tropopause heights, using data that is specific to the time and location of the VMR and aerosol extinction measurements of interest is important. Concurrent temperature profiles are retrieved alongside VMR profiles from ACE; specifically, ACE temperature profiles are determined from 18 km upwards from data retrievals while temperatures and pressures below 18 km are fixed to data from the Canadian Meteorological Centre weather model (Sica et al., 2008). Therefore, a temperature-based tropopause definition, the WMO lapse rate tropopause, was used in this study (WMO, 1957).

In practice, the lapse rate tropopause was calculated by first determining the lapse rate of the ACE temperature profile at every vertical level, and interpolating to determine slope values between the 1 km vertical intervals. Every vertical level where the lapse rate reached $-2$ K km$^{-1}$ was marked, and the lowest level at which the lapse rate passed this threshold was determined to be the tropopause altitude. For temperature profiles that exhibit a double tropopause, this method provides the lower tropopause altitude (Peevey et al., 2012; Homeyer et al., 2014).

## 3 Results and discussion

## 3.1 Chemical signatures for biomass burning

The tropopause acts as a strong barrier to troposphere-to-stratosphere transport, and the high vertical resolution of ACE is useful in determining whether smoke entered the stratosphere. Individual occultation measurements from ACE provide simultaneous constituent volume mixing ratio (VMR) and temperature profiles where biomass burning product VMRs can be compared against self-consistent tropopause heights as measured on the same occultation and in broader averages. HCN is a robust wildfire tracer for the Northern hemisphere because its primary source is biomass burning (Li et al., 2000; Roberts et al., 2020). In contrast, other tracers like CO and ethane are emitted in large amounts by both wildfires and anthropogenic sources such as transportation and industry (Xiao et al., 2004), complicating source attribution for those compounds.

Additionally, wildfires are not the only source of particles in the stratosphere, especially in the recent context of the Hunga Tonga–Hunga Ha'apai and Shiveluch eruptions in the last couple of years. This is why we focus on chemical data from ACE–FTS in addition to aerosol data to be able to confidently attribute wildfires as the source.

We first investigate average monthly HCN volume mixing ratio (VMR) profiles. Some individual occultations do not extend into the troposphere, so monthly averages in absolute altitude capture the maximum amount of data to compare 2023 with the rest of the ACE-FTS measurement period (2004 to 2022). A tropopause-relative framework is used in Sect. 3.2 to detect individual measurements of stratospheric smoke. Chemical anomalies detected in the region from 40° to 70° that are likely associated with wildfires also reflect other active burning regions such as Siberia (MODIS, https://modis.gsfc.nasa.gov/).

Figure 1 suggests an enhancement of HCN in the upper troposphere and lowermost stratosphere (UTLS) between June and September 2023 relative to preceding years, with the monthly average tropopause heights calculated from ACE-FTS temperature profiles representing a rough delineation between the troposphere and stratosphere. On average, enhancements do not extend more than a couple kilometers above the tropopause. Similar results can be seen for single profiles (Fig. S3).

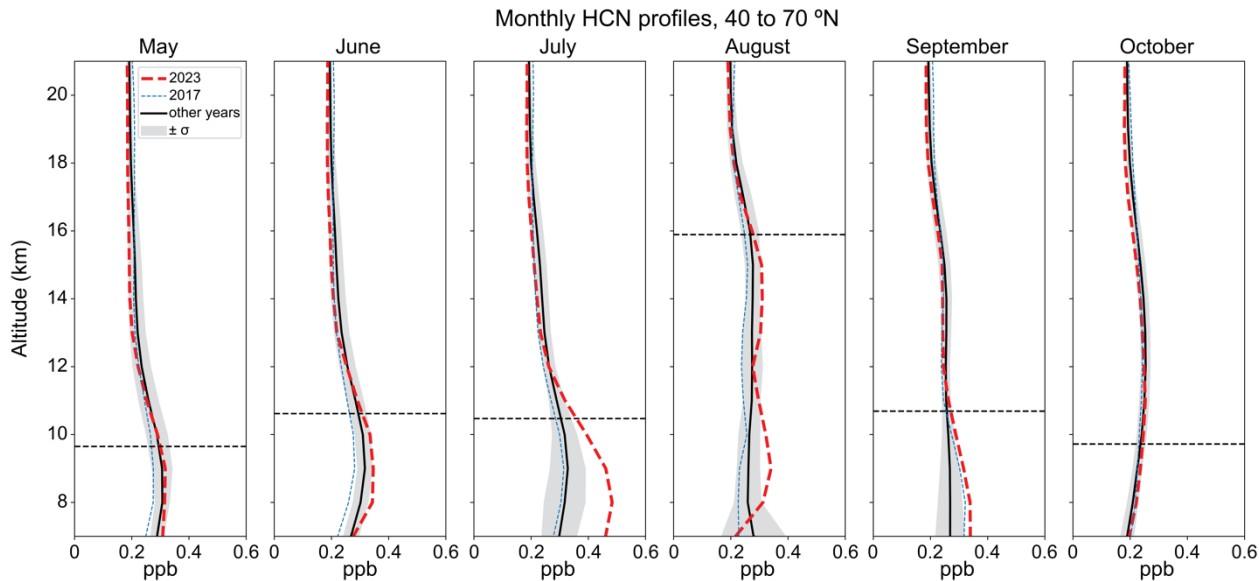

**Figure 1: Monthly average HCN profiles from 40 to 70° N measured by ACE-FTS. The red curves show monthly averages in 2023,**
**and the black curves show monthly averages from 2004–2022 with the grey shaded area representing ± one standard deviation from the 2004–2022 average. The blue dashed line indicates 2017 averages as it was an anomalous year with the PNE. The monthly average lapse rate tropopause altitudes calculated for 2023 are plotted as horizontal dashed lines for reference.**

ACE/SCI-SAT1 takes a very limited number of measurements over our latitude range of interest during August as seen in Table S1; only 20 measurements were taken in August 2023, and all on the first couple of days of the month and between 40
and 45° N. This may explain the unusually high tropopause altitude for this month as tropopause altitudes are higher closer to the equator and previous studies with ACE in August also report a higher tropopause height (Doeringer et al., 2012). Additionally, a low number of measurements are likely more susceptible to transient meteorological influences such as the

Asian monsoon (Basha et al., 2020). ERA5 reanalysis data collocated at ACE measurement points and times for August yield a similarly high average tropopause height of 14.6 km (Herschbach et al., 2023).


The range given by the standard deviation of ACE data from 2004 to 2022 indicates that there were likely other wildfire years that also produced high HCN mixing ratios from June to September, but 2023 is on the upper end of this spread from about 8 to 11 km. July 2023 in particular has noticeably higher abundances compared to the range of previous years. Although this HCN enhancement is indicative of wildfire smoke reaching the upper troposphere, the occurrence of

stratospheric injection is less clear given the variability of tropopause heights and the limited number of observations, particularly in August (see Table S1). This limited coverage and amount of August data ACE is supplemented by the higher coverage of OMPS as shown in the following section and MLS in the supplemental information.

The average July 2023 profiles of other biomass burning markers such as CO, $CH_3OH$, HCOOH, and $C_2H_6$, also exhibit

elevated VMRs in the upper troposphere but not significantly so in the lowermost stratosphere (Fig. 2).

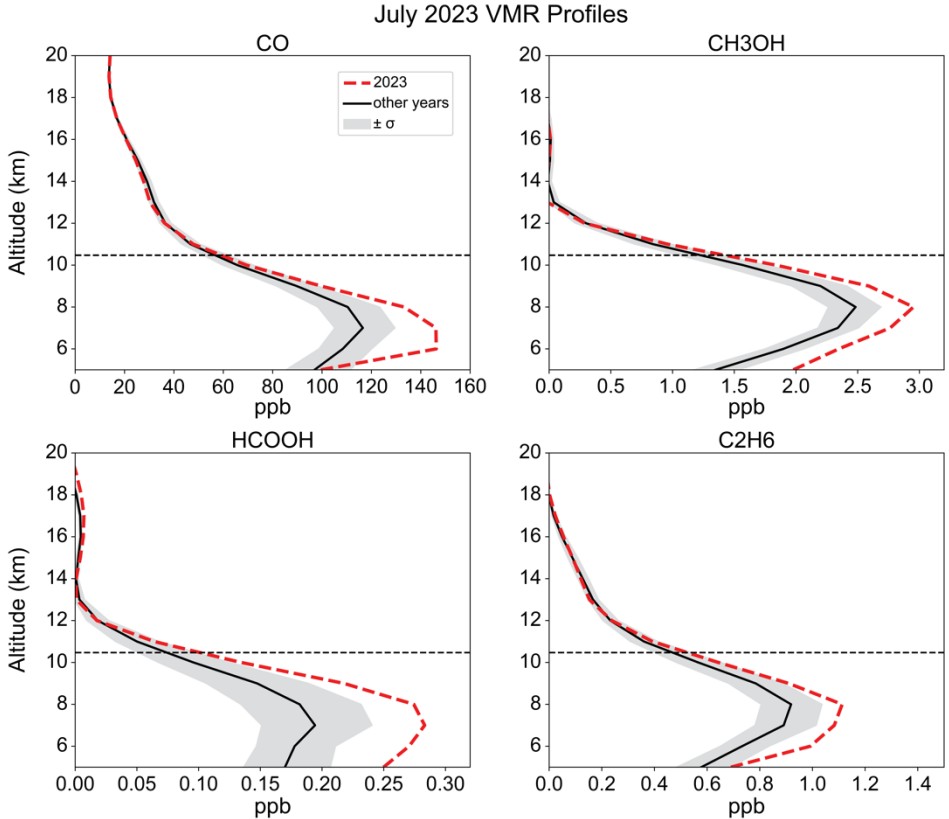

**Figure 2: Average July 2023 VMR profiles for additional biomass burning tracers: CO, $CH_3OH$, HCOOH, and $C_2H_6$, plotted against 2004–2022 July averages over 40° N to 70° N. The grey shaded area represents ± one standard deviation based on 2004–2022 data.**

These data provide strong evidence for the vertical transport of wildfire smoke to the upper troposphere due to high pyroCb activity during the summer wildfire season, which can influence upper tropospheric composition and chemistry (e.g. ozone production). However, significant stratospheric penetration is not observed in the averages.

## 3.2 Stratospheric smoke signatures

Inspection of every individual occultation measured by ACE-FTS over the 40 to 70º N latitude band in the 2023 burning
season revealed a few occultations indicative of stratospheric smoke, exhibited by enhanced chemical signatures and aerosol extinction above the tropopause. These measurements offer evidence of a small amount of smoke in the lowermost stratosphere. The July occultations that exhibit multiple biomass burning products and aerosol extinction measured in the stratosphere are shown in Fig. 3, and similar profiles for other months are shown in Figure S5.

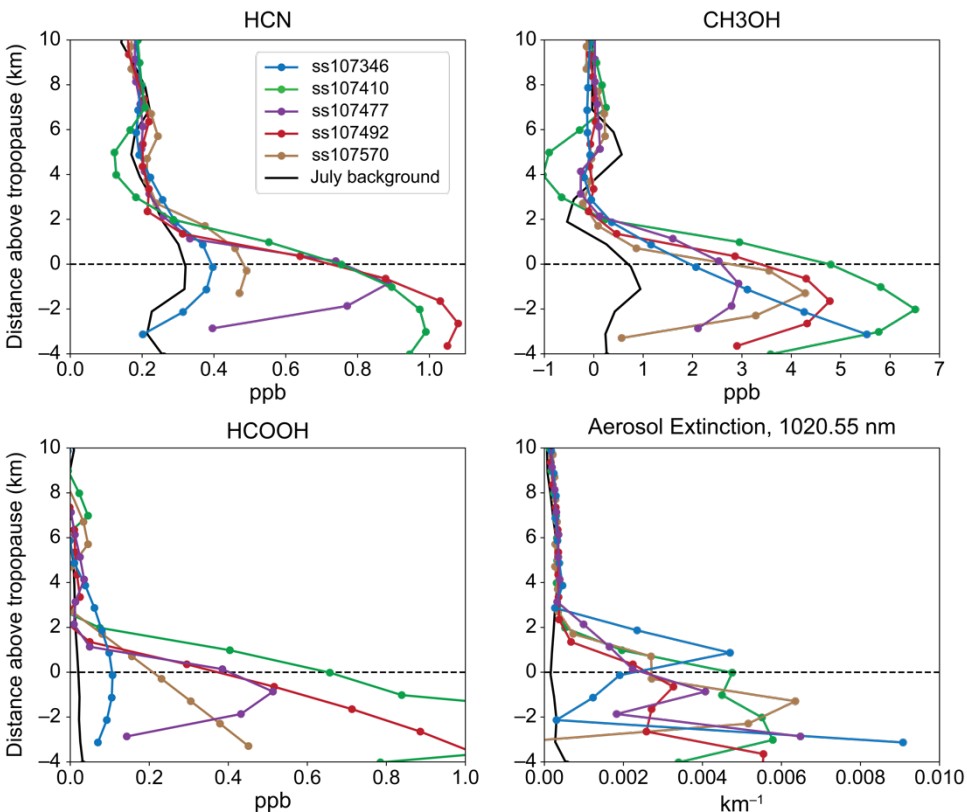

**Figure 3: Tropopause-relative profiles for July 2023 occultations that exhibit enhanced wildfire product VMRs and aerosol**
**extinction in the lower stratosphere. A background profile with no smoke from a similar location in July 2023, ss107249, is plotted for reference.**

Elevated VMRs of multiple biomass burning products measured above the tropopause in these data offer evidence for a nominal amount of wildfire smoke in the stratosphere. To further validate this conclusion, the ACE infrared absorption

spectra were also analyzed for features characteristic of wildfire smoke, as in Boone et al. (2020). Carbonaceous aerosols
typically exhibit C–H, O–H, and C–C features associated with alkanes and oxygenated organics, and these are indeed present
for these occultations, for example as shown in the IR spectrum in Fig. 4 (Zhong and Jang, 2014; Boone et al. 2020; Bernath,
2020). The O-H stretch is commonly seen in smoke particles and indicates that they are oxygenated (Boone et al., 2020). The
C=O stretch, which is also associated with oxygenated smoke particles and identified using a feature around 1750 cm$^{-1}$,
cannot be seen because that region is saturated at lower altitudes by strong absorption of water vapor in the same spectral
region.

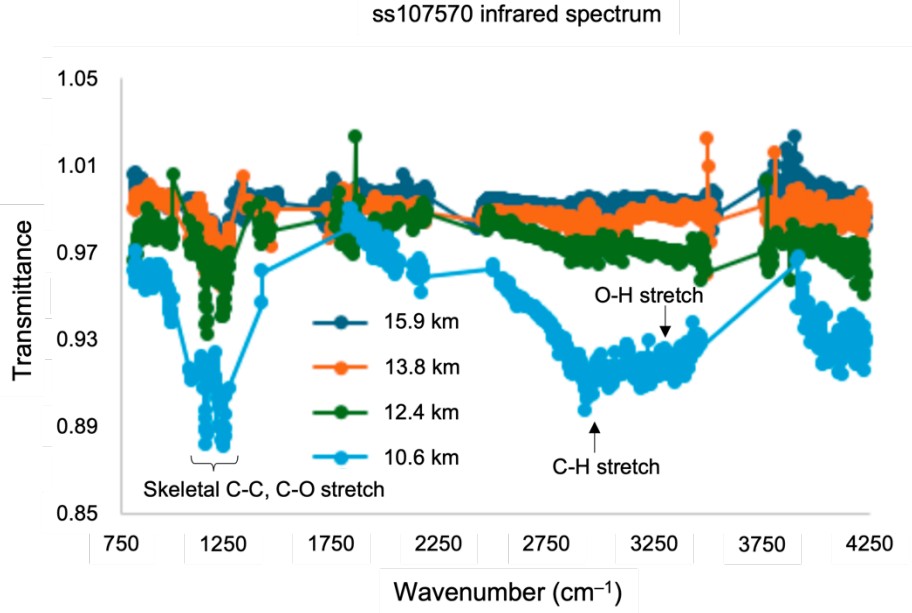

**Figure 4: Residual IR spectrum for ss107570 measured at multiple tangent heights. The highlighted features at 10.6 km are indicative of smoke. At higher altitudes, the spectrum becomes plume-free and there are no longer indicators of smoke.**

Given the extent of Canada's wildfire season, using fire databases to identify potential source fires is non-trivial given the large number of events in 2023 with extensive burned areas (Canadian Interagency Forest Fire Centre Inc, https://ciffc.net/national). Thus, more targeted methods for identifying pyroCb-specific fires are a more promising approach for this analysis. Cross-referencing with an online crowdsourced pyroCb database (https://groups.io/g/pyrocb) suggests a few pyroCb events with brightness temperatures below –55 ºC. A cold cloud top of this magnitude is indicative of deep overshooting convection that may penetrate the tropopause and enter the stratosphere (Romps et al., 2009). It is therefore plausible that these pyroCbs injected smoke into the stratosphere.

Back trajectories initialized from ACE stratospheric smoke measurements on NOAA HYSPLIT with GDAS 1º meteorological data do not directly intercept any of these reported pyroCbs, but this is not surprising given the limited spatial coverage of ACE. Trajectories initialized from two occultations, ss107346 and ss107570, pass within tens of

kilometers and a few hours of reported pyroCbs and are shown in Fig. S6. Despite the lack of direct detection, chemical
signatures of smoke after dispersion in the lower stratosphere are clearly measured and show the limited vertical range of
wildfire influence in the stratosphere: within 2 km above the tropopause.

### 3.3 OMPS LP Aerosol data

Aerosol extinction data from OMPS LP also indicate a minor increase in aerosol burden in the lower stratosphere averaged
between 11.5 and 16.5 km, in late July 2023 (Fig. 5). However, this increase above background levels is both small in
magnitude and short-lived, which indicates that not enough smoke was injected into the stratosphere to significantly impact
extinction measurements. Throughout the entire wildfire season, there are no significant aerosol extinction signals at this
altitude range that would suggest deep convective wildfire events. This validates our general finding that very little smoke
managed to make it well above the tropopause despite high pyroCb activity, suggesting that the many convective events that
occurred were mostly limited to the upper troposphere. There is a constant background level of aerosols above 60° N, but
this feature is present from the beginning of 2023 and therefore is not linked to the 2023 Canadian wildfire events but rather
other aerosol sources such as volcanic eruptions.

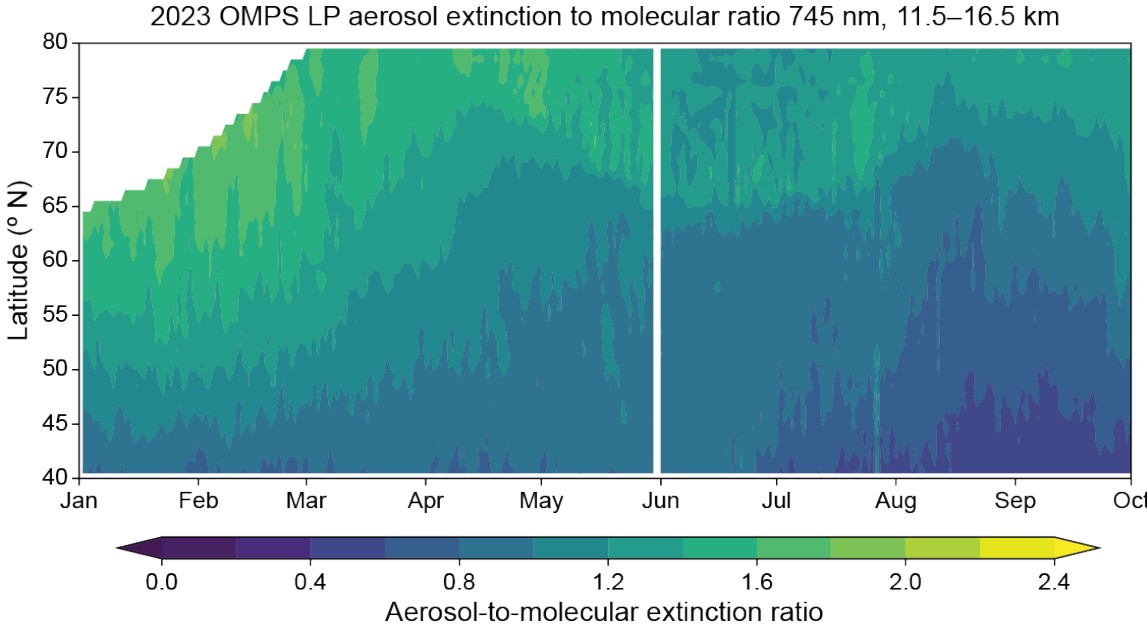

**Figure 5: Zonally averaged daily OMPS extinction to molecular ratio between 11.5 and 16.5 km, 40 to 80° N. This represents the
evolution of average aerosol load in the lower stratosphere during the 2023 summer wildfires.**

The altitude range between 10.5 and 11.5 km contains either tropospheric or stratospheric air depending on the latitude and
time of year, and analysis of OMPS data at this altitude reveals substantial average aerosol loading in the region around the
tropopause in 2023 (Fig. 8, middle panel). This suggests that the many pyroCbs of the 2023 wildfire season alongside other

aerosol sources managed to inject particles right around the tropopause. However, the significantly lower aerosol extinction signal past 11.5 km as seen in Fig. 5 and Fig. S7 constrains these inputs to the region at or just above the tropopause as opposed to higher in the stratosphere where more impacts would be realized.

## 3.4 Comparison to Pacific Northwest Event

To contextualize the 2023 Canadian events, we compare its ACE-FTS profiles with measurements of the 2017 PNE which also occurred in western Canada during the summer. Analysis of this event was previously reported in Boone et al. 2020, with the occultation sr75758 probing the plume a few weeks after initial stratospheric injection. Figure 6 shows a substantial difference in smoke altitude, with the 2017 PNE leading to chemical signatures of biomass burning products measured above 20 km compared to the much lower vertical extent of the 2023 fires.

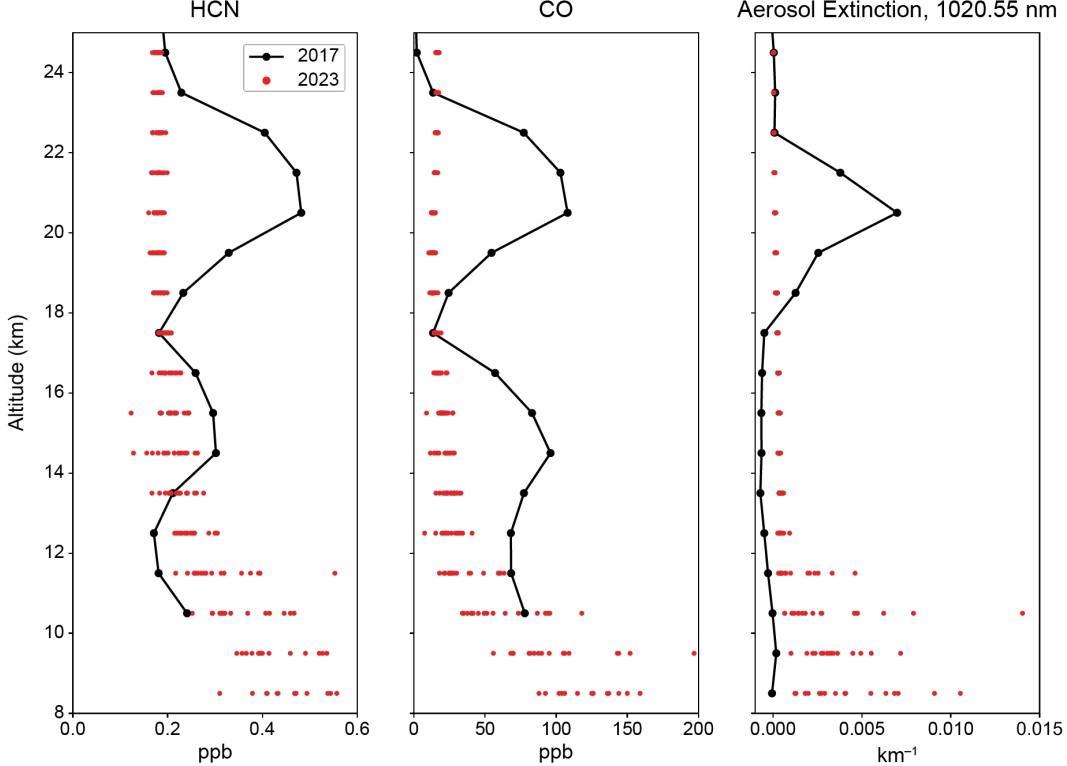

**Figure 6: Biomass burning product VMR and aerosol extinction profiles for occultation sr75758 (black) associated with the 2017 Pacific Northwest Event (Boone et al., 2020). Profiles from the 2023 Canadian wildfires that exhibited stratospheric penetration are shown in red for comparison.**

The top panel of Fig. 7 also shows with OMPS data that 2017 featured a much larger injection of aerosols between 11.5 and 16.5 km relative to 2023, where there is no evident increase in average stratospheric aerosol extinction throughout the entire summer. Similarly, the bottom panel of Fig. 7 contrasted to Fig. 5 shows that the magnitude and duration of stratospheric perturbation from the PNE was much more substantial than that of any pyroCb activity from 2023. Since ACE-FTS

measured only a handful of occultations with stratospheric signatures of smoke and the OMPS LP aerosol data does not feature any significant perturbations in 2023 past 12 km, we conclude that the 2023 Canadian wildfire season did not significantly impact stratospheric composition relative to previous years despite burning a far larger area and exhibiting frequent pyroCb activity.

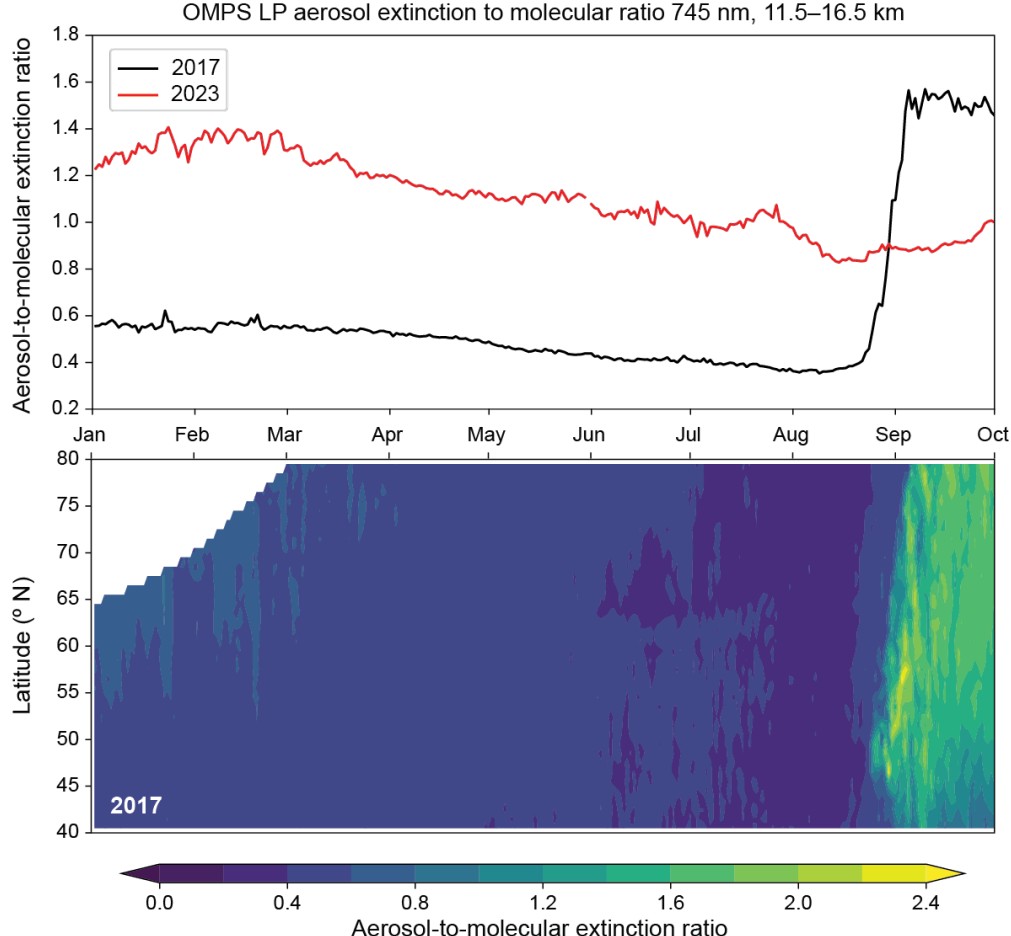

**Figure 7: Top panel: average OMPS aerosol-to-molecular extinction ratio from 40 and 80º N, 11.5 to 16.5 km compared between 2017 (black) and 2023 (red). Bottom panel: 2017 zonally averaged OMPS extinction between 11.5 and 16.5 km, 40 to 80º N.**

However, the aerosol loading near the tropopause is similar between the two years, as seen in Figure 8. It is clear that the active pyroCb activity in the 2023 wildfire season did influence upper tropospheric composition given the increase in aerosol extinction starting in May, which coincides with the start of the burning season. This is in stark contrast to the stratospheric aerosol loading from Figure 5, which is largest at the beginning of the year. Thus, the stratospheric impact of the 2017 PNE is visibly more significant since smoke following the event is measured well above the tropopause, whereas smoke from the 2023 fires remains around the tropopause throughout the entire wildfire season. This lower altitude of perturbation

245 corresponds to a shorter smoke lifetime in the stratosphere, further suggesting the lesser impact of the 2023 fires compared to the PNE (D'Angelo et al., 2022).

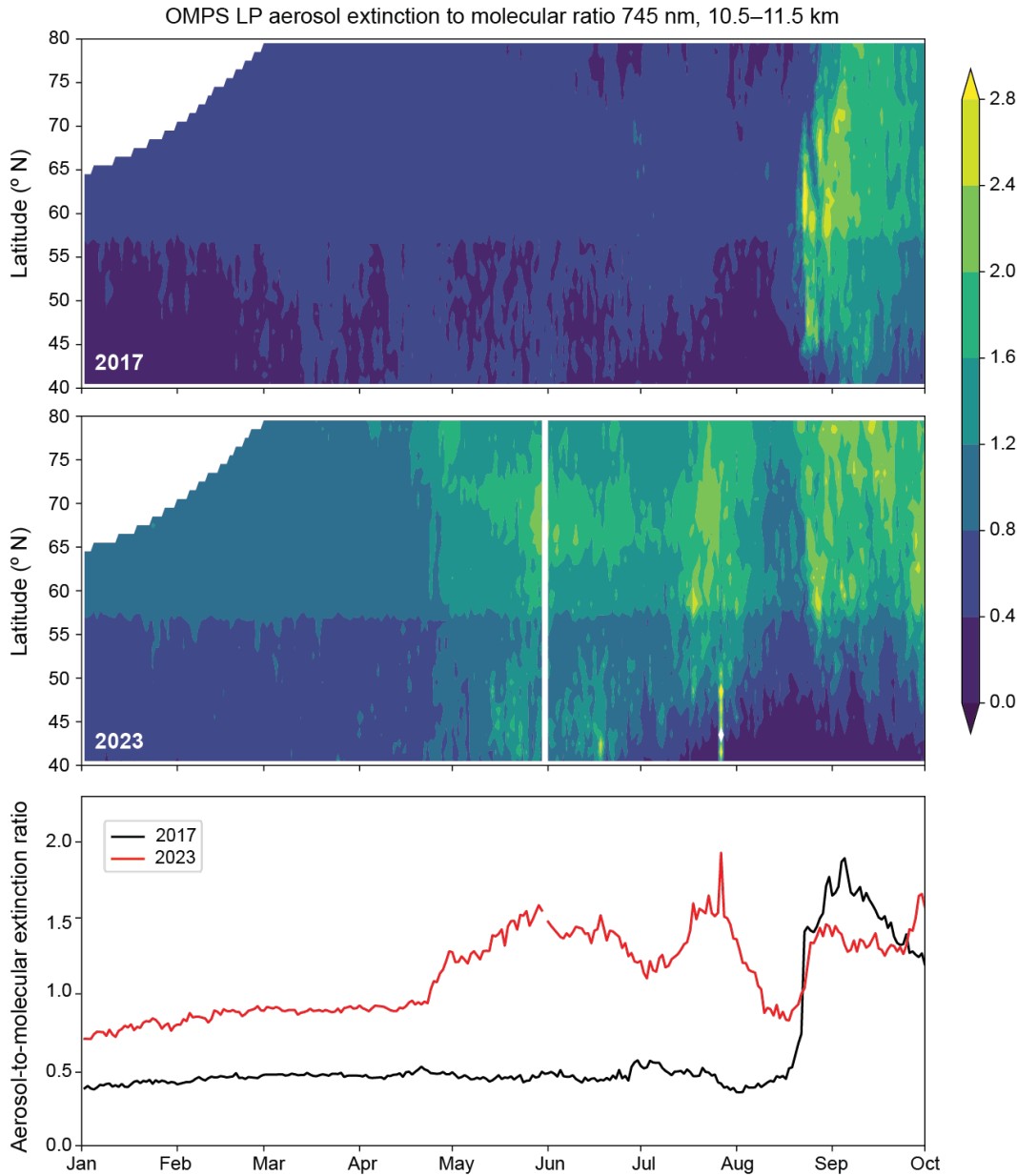

**Figure 8: Top panel: 2017 zonally averaged OMPS aerosol-to-molecular extinction ratio between 10.5 and 11.5 km, 40 to 80º N.**
250 **Middle panel: 2023 zonally averaged OMPS aerosol-to-molecular extinction ratio between 10.5 and 11.5 km, 40 to 80º N. Bottom panel: average OMPS aerosol-to-molecular extinction ratio from 40 to 80º N, 10.5 to 11.5 km compared between 2017 (black) and 2023 (red).**

## 4 Conclusions

Using ACE-FTS observations, we identified the presence of biomass burning products in the stratosphere in a limited number of occultation measurements over Canada during the 2023 summer wildfire season. This and ACE imager data, alongside OMPS LP aerosol data and MLS CO data, suggests that the immediate impacts of these fires are essentially limited to the troposphere, with evidence only for nominal amounts of smoke in the lowermost stratosphere. Any aerosols that made it into the stratosphere remained near the tropopause and did not make it high enough to substantially influence stratospheric composition. However, carbonaceous aerosol has the potential to increase parcel buoyancy via solar heating and self-loft over longer timescales (de Laat et al., 2012; Yu et al., 2019; Ohneiser et al., 2023). The impacts of this process have not been pursued in this work but may have implications for climate and stratospheric composition. Additionally, the climate implications of increased smoke in the upper troposphere is an active area of interest (Christian et al., 2019; Kochaniski et al., 2019; Li et al., 2021).

In summary, this work shows that despite an extremely extensive wildfire season with frequent pyroCb activity in Canada, the conditions for sufficiently deep convection were met so rarely and to such a limited extent that no significant stratospheric perturbation took place. These results highlight that area alone is not a useful indicator of the potential for stratospheric effects, and projected increases in area burned per degree of global warming are not enough to forecast the vertical extents of future wildfires. The complexity of intense pyroCb formation motivates further study of why certain events, such as the 2019 – 2020 ANYSO and 2017 PNE, are more primed for stratospheric penetration. Factors including fire intensity and atmospheric structure may play an important role. Understanding which wildfire conditions enable stratospheric impacts, and how and where these may be realized under a changing climate, has significant implications for stratospheric ozone and climate.

## Competing interests

The contact author has declared that none of the authors has any competing interests.

## Acknowledgments

Selena Zhang was funded by the MIT John H. Olsen Presidential Fellowship. Susan Solomon gratefully acknowledges support by NSF-AGS grant 2316980. Funding for the Atmospheric Chemistry Experiment is provided by the Canadian Space Agency.

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
