# Peer review of "Investigating the vertical extent of the 2023 summer Canadian wildfire impacts with satellite observations"

_EGUsphere, 2024_

## Referee Comment (RC1)

Review of Zhang et al., "Investigating the vertical extent of the 2023 summer Canadian wildfire impacts with satellite observations".

Reviewer: Mike Fromm

Zhang et al. use satellite remote sensing observations to evaluate an attention-getting atmospheric condition in the northern wildfire season of 2023. During that season, fires across the extratropical globe—especially in Canada—were unusually numerous, large, and active. The fire season was punctuated by an unprecedented number of pyrocumulonimbus (pyroCb) storms, again particularly in Canada. A natural science question arises: How did these fires and pyroCbs perturb the upper troposphere and lower stratosphere (UTLS) compared to other years that experienced large stratospheric smoke perturbations? The hypothesis might be an expectation of an unusually large and/or persistent biomass burning plume in the 2023 stratosphere. Zhang et al. tackle that question and hypothesis. This is self evidently an appropriate topic for ACP.

Zhang et al. use ACE-FTS, OMPS-LP, and MLS data in their pursuit. All three are well suited, individually and in combination, to this study. Indeed, they may be optimal as compared to, for instance, models or localized observation systems. The authors are commended for assembling these data in short order after the 2023 season. I would expect that a thorough deployment and analysis of these data could address the above-mentioned science question.

Whereas the authors have made available an interesting set of observations, the current presentation generated—in my assessment—more questions than answers. These concerns are listed in detail below, first the substantial issues, then minor/technical ones. The list is fairly long, which represents to my mind a systemically unconvincing line of reasoning and/or presentation.

Here is a summary of my concerns. The authors report that only a single occultation profile from ACE-FTS embodies indicators of stratospheric biomass burning emissions in the stratosphere in 2023. If true, this would indeed be a notable finding, given the ~800 profiles in play. But I was left doubtful by the way the data were presented. The authors identified MLS as a core data item but presented only a limited analysis rather than a season-long survey. The authors' deployment of OMPS-LP aerosol data was not thoroughly described and the presented results were confusing, perhaps even in error. Air-parcel trajectories were used to connect a single ACE stratospheric observation to a potential pyroCb, but that analysis was unconvincing.

The potential is there for a substantial improvement by addressing the concerns detailed below. The author team is ideally equipped to make the necessary improvements. My hope is that my report is accurate and helps guide the authors to the key areas that need revision before this manuscript merits publication in ACP.

Below I list all the issues in bold text to guide the authors to line numbers, quotes, and/or figures. My input follows in plain text.

**Substantial Concerns**

**Line 80:** The authors give two reasons for invoking MLS:
1. ACE's relatively poor spatial coverage
2. as a validation of ACE CO.
But they only show a single plot in Supplemental Information. As far as I can tell from the literature, ACE needs no validation of its CO retrievals. Moreover, MLS data aren't exploited for its superior coverage. I see little to no value in the MLS component of this manuscript. My recommendation is to more fully analyze MLS data for the 2023 season or drop this part of the work.

**Line 89:** This contention needs more explanation. MLS CO is good down to 215 hPa, which is nearly ideal for this study. MLS 215 hPa CO was used quantitatively in the study of the PNE smoke event (https://doi.org/10.5194/acp-21-16645-2021), Black Saturday and Great Divide (doi:10.5194/acp-11-6285-2011). What is the basis for questioning some "**individual profiles**"? The broader MLS vertical resolution would seem to be well compensated for by its twice daily sampling, near global orbital coverage, and high-resolution sampling along orbit.

**Line 132; Figure 1:** I'm perplexed by the high August tropopause and the explanation. The twenty August ACE occultations are distributed evenly, longitudinally. For an average tropopause to be 16 km, some individual ones would have to be even higher to compensate for those distant from the Asian Monsoon sector (which represents at most about ¼ of the zonal belt even at its widest, south of 40N). I examined the August 2023 ACE occultations north of 40N and the temperature data. Eyeballing the tropopause from the temperature profile, I estimated an average ~14 km. I also used NCEP Reanalysis data, interpolated in space and time to the ACE occultations, to calculate tropopause height based on the dynamical definition. The average for these occultations was 13.1 km using the threshold: pv=2.5 pvu. If my results are accurate, this disparity calls into question tropopause height calculations here and perhaps elsewhere in the paper. Considering the crucial importance of the tropopause height in this study, I would encourage the authors to check their calculation method and results, and update all the analyses dependent thereon.

**Line 132, Figure 1:** The August panel is also puzzling in that the 2017 and ACE-lifetime statistical HCN are larger at 16 km w.r.t. the other months. Even the sigma width is larger. Are the August data from other years all subject to a similar tropopause bulge? Is the entire ACE August 40-70N record that much different than the adjacent months? Because the August pattern in Figure 1 is so strikingly different, it needs to be addressed exhaustively or removed from the analysis.

**Line 138:** This is an argument for using a tropopause-relative reference frame. Given the physical reality of the 2023 plume heights congregating near the tropopause, it is recommended that a tropopause-relative analysis be performed as a replacement or complimentary construct.

**Line 138:** A possible complicating factor comes to mind while contemplating this analysis. The phenomenon of the double tropopause [e.g. Homeyer et al. (2014; https://doi.org/10.1002/2014JD021485)] could well be a factor in the tropopause-height calculation and assessing 2023 plume tropopause-relative height. It might be worth noting this and expressly stating if/how double tropopause situations are handled in the calculation of tropopause height.

**Figure 3:** These statistics show that the authors have done a trop-relative analysis of all ACE data. So, it should be considered elsewhere.

**Figure 4:** It would be helpful to the broad readership to add a plume-free spectrum for comparison, akin to Figure S4 in Boone et al. (2020).

**Line 163, "saturated":** Saturated with what, or because of what? This needs to be fleshed out because 3 of three distinct smoke plumes shown in Boone et al. (2020) all had a prominent C=O stretch feature. So, the disparity of the present example w.r.t. these published examples raises the question of why it is different. IR spectroscopy of smoke as compared to other particulate compositions is not yet a mature and intuitive topic for the general science audience. Help is required for the benefit of those not attuned to the specifics of this niche ACE-FTS data application.

**3.2 Occultation ss107570:** This section is where the authors state that only one occultation (ss107570) had a demonstrable smoke enhancement above the tropopause. I have examined all the 2023 ACE profiles north of 40N, in concert with tropopause heights, and found several that are as convincing as ss107570. E.g. ss107346 and sr108266. Given the above-mentioned concern with the tropopause-height calculation, I encourage the authors to re-examine the occultation record for other examples of extinction enhancements clearly above the tropopause to find other candidates for stratospheric smoke. This would have two crucial benefits, 1. Additional IR spectra to exploit, and 2. Other candidates for matching the ACE observation to a potential pyroCb. Another suggestion to the authors is to extend their analysis to October ACE data, considering the fact that there were two Canada pyroCb events late in September and the likelihood of ACE stratospheric smoke observations post September.

**Section 3.3:** Either this section or section 2.2 need to contain a detailed description of the OMPS analysis method. Much is unknown. How are clouds handled? How is the vertical data ensemble between 11-16 km treated and reduced to single extinction ratio values? How is the molecular extinction calculated? What is the publication background on this method, if any?

**Figure 5:** The color range here seems to indicate an average extinction ratio <<1.0. This plot does not conform to Figure 8 (top). The general average between 40-80N is <<1.0 in Figure 5 while Figure 8 shows a baseline near 1.0. Something appears to be amiss.

**Section 3.4:** Suggestion: Move this section to 3.3 to connect it directly to the individual ACE profile presented in 3.2.

**Figure 6:** I don't understand the rationale for launching 3-day trajectories within a 3.5-day span of launch times. How does this **"compensate for limited ACE-FTS coverage?"** The ACE occultation location and time are known precisely. The back trajectory from ACE time does not nearly extend long enough to implicate any land/fire surface area. Only the trajectories launched ~3 days before ACE pass over a suspect landscape. If one were to launch forward trajectories from these times to ACE time, it is not certain where they'd end up and if/how they would apply to the ACE occultation. Hence, as presented, this analysis is unconvincing. It is suggested that the authors restrict launch times to ACE time and run them back long enough to encounter locations/times of reported pyroCbs.

**Figure S4:** Ditto the concerns expressed above. In this case there is some wiggle room for initiation time, but only on the order of a couple hours. As it stands, the trajectory launched at pyroCb time ends up somewhat close to the ACE location but 3 days before the ACE occultation.

**Line 218, "The maximum VMR and aerosol extinction values are similar between the two events, and Fig. S5 also shows that the aerosol loading near the tropopause is similar between the two years.":** This sentence is confusing. Are the data in Fig. 7 taken as representative of PNE and 2023, such that the comparative gas and aerosol data maxima reflect a similarity between the PNE and 2023 **"events"**? Moreover, the PNE profile's CO max is not similar to the 30 July 2023 CO max. The extinction max for the two are similar, but the value for 2023 is in the troposphere. Substantial elaboration and clarification are called for. This is especially important given that the authors relate their synthesis of Figure 7 to the season-long graphic in Figure S5.

**Line 219-221, "However, the stratospheric impact of the PNE is visibly larger since the entire plume is measured well above the tropopause whereas only part of the Yukon plume is clearly in the stratosphere.":** Again, the authors seem to be taking this singular PNE profile as somehow representative of PNE in general, and in terms of injection height specifically. The PNE pyroCbs did not inject smoke to the tropopause+9 km.

**Figure 8:** The pre-PNE color shows extinction ratio <= 0.0 according to the color bar, but the line on the upper panel indicates values exceeding 0.4. Is there a discrepancy here?

**Figure 8:** The difference in extinction ratio between 2017 and 2023 is striking, from the beginning of May onward. The loading in 2023 is 2 to 3 times larger than 2017. What explains this difference? The earliest Canada pyroCbs were on 4-5 May, so the large 2023 values from onset are difficult to understand. What was the extinction ratio pattern prior to May? Do the authors have an explanation for this apparent puzzle?

**Figure 8:** The 2017 extinction ratio jumps to values ~1.5 after PNE in the lower panel. But the color-scaled top panel manifests no such value, even at its maximum value. Something is apparently amiss.

**Figure S1:** If this is to remain in the paper, as a validation of ACE CO, more information is called for. Was there an attempt to match MLS profiles with the ACE profiles? Were all the MLS data, day and night, included in the averaging? If essentially all the MLS data 40-70N are used, it probably doesn't qualify this as an ACE validation.

**Figure S2:** Recognizing the authors' uncertainty about enhanced HCN w/o extinction, certain data points call out for an explanation. By that I mean the gray enhancements, high above the tropopause, exceeding the red-dot enhancements in May, July, August, and September. As such, the reader might wonder about how robust the red dots are or what to make of these gray HCN enhancements. Please explicitly deal with these perplexing data points.

**Figure S5.** The top two panels are defined as extinction ratios for a single, thin layer between 10.5 and 11.5 km. But the bottom panel is defined as relating to 11.5-16.5 km (like the plots in the main body). Is this the authors' intention? It seems odd.

**Figure S5 (bottom panel):** The 2023 time series shows a pronounced mound of extinction ratio from mid-July to August.  This is not apparent in Figure 5, which is this panel's mate, according to the figure caption. It looks much more like the 2023 panel above. Regardless of which panels are matches, the numeric values in the time series do not conform to the color-scaled plots in either figure. Something is apparently amiss.

**Figure S5 (bottom panel):** How is this panel's construction for 2017 different than Figure 8's top panel? Both are described identically in their captions but the time series lines are different.

**Figure S5:** Logically, this figure—once corrected--belongs in the main paper. It is central to the authors' thesis.

**Technical Issues**

**Abstract, Line 29:** Mention MLS here given that ACE-FTS and OMPS-LP are mentioned.

**Line 18:** Replace "**Profile**" with "Profiler."

**Several locations:** use hectares instead of acres.

**Line 51, "It has been reported that at least 135 pyroCbs occurred in Canada between May and August 2023…":** Please cite the "report."

**Line 53, "…perturbations to stratospheric composition (NASA EarthData, https://www.earthdata.nasa.gov/).":** It's not clear what this citation refers to in the context of this sentence. This is a generic data web site. Please provide a more specific citation.

**Line 81:** A citation is needed for the MLS instrument.

**Line 127, "2017…":** Please reword to avoid starting a sentence with a numeric.

**Figure 1 caption, "…tropopause altitudes calculated for 2023 are plotted for reference.":** Please insert **"**as horizontal dashed lines" after "**plotted**".

**Figure 1 caption, "…approximate monthly average…":** What is meant by the "**approximate**" qualifier? Please consider dropping this unless there is a purpose for it.

**Line 133, "external processes":** What is meant by "**external**" here? What processes would be internal? Please clarify or reword.

**Line 137 and elsewhere, "concentrations":** Mixing ratio instead? ACE retrieves mixing ratio, which is a cousin to concentration, but they are distinct. Please clarify.

**Figure 2:** Are the horizontal dashed lines tropopause height? These are ~0.5 km different than the July value in Figure 1. Please clarify or correct.

**Figure 3 ordinate titling:** Replace "**above**" with "relative to the".

**Figure 3, lower right panel:** Please consider adding statistical results as shown in the other 3 panels.

**Line 185, "72 hour trajectories…":** Please reword to avoid starting a sentence with a numeric.

**Line 187, "shown in Fig. 6, with possible source fire locations":** How are the source fire locations shown in FIg. 6? I don't see any marks or symbols, or description in the caption.

**Figure 6 caption:** Please correct the ACE longitude.

**Line 195:** Change "**Fig. 5**" to "Fig. 6".

**Line 204:** Delete "mean".

**Line 214, "…~1 km injection height…":** Please correct the number.

**Figure S3 caption, "Atmospheric profile…":** Please replace "**atmospheric**" with "temperature."

**Figure S3**: Please provide an explanation of the data set used for this temperature profile. Since it is stated that the data are at 03 UTC, it is unlikely that the profile is sourced in radiosonde data.

---

## Author Comment (AC1)

We would like to thank the editor for handling the review process for our manuscript, and the referees for reviewing the manuscript. We have modified our manuscript in response to the referee suggestions. Our responses are listed below; comments from the reviewers are in plain text, and our replies are in **bolded blue text**. The line numbers in our responses refer to line numbers in the revised manuscript.

**All of the comments and responses are listed below, but first we highlight some common comments across reviewers and how these were addressed. For example, additional MLS analysis with its greater coverage as noted by the reviewer was added to the supplemental information to complement our conclusions from ACE-FTS and OMPS LP. Further description of the tropopause calculation and the anomalously high August tropopause were added. The trajectories from ACE occultations were also changed to cover a longer time range (315 hours) and are now initiated at the time of ACE measurement as suggested.**

Referee 1 (RC1):

Line 80: The authors give two reasons for invoking MLS:
1. ACE's relatively poor spatial coverage
2. as a validation of ACE CO.
But they only show a single plot in Supplemental Information. As far as I can tell from the literature, ACE needs no validation of its CO retrievals. Moreover, MLS data aren't exploited for its superior coverage. I see little to no value in the MLS component of this manuscript. My recommendation is to more fully analyze MLS data for the 2023 season or drop this part of the work.

**Response: Thank you for this helpful point. We wanted to show that two different instruments with varying spatial and temporal coverages provided similar profiles in a monthly averaged sense, which supports the view that over longer time scales as the smoke is mixed and transported, they are comparable despite sampling differences. It is true that ACE does not need validation on its measurements and this was not the purpose of the comparison. In addition to this profile comparison, we have now added an analysis of MLS CO data to the supplemental information in Figure S1 (as described further in the following response).**

Line 89: This contention needs more explanation. MLS CO is good down to 215 hPa, which is nearly ideal for this study. MLS 215 hPa CO was used quantitatively in the study of the PNE smoke event (https://doi.org/10.5194/acp-21-16645-2021), Black Saturday and Great Divide
(doi:10.5194/acp-11-6285-2011). What is the basis for questioning some "individual profiles"?
The broader MLS vertical resolution would seem to be well compensated for by its twice daily sampling, near global orbital coverage, and high-resolution sampling along orbit.

**Response: Yes we agree; as you state, coverage from MLS is _nearly_ ideal. However, we are interested in determining stratospheric penetration and 215 hPa is on the cusp of being satisfactory for our analysis. Since the data product is not recommended for use below 215 hPa, it is difficult to assess the vertical profile of CO beyond the lower stratosphere which is why it is not a central part of our analysis.**

**We add this point in line 85-87:**

"Since we are interested in the vertical profiles of biomass burning products, MLS is not as useful as ACE for our analysis given the lack of reliable data at our pressure range of interest in the lowermost stratosphere and its transition to the tropopause and upper troposphere."

**Despite this, MLS certainly provides useful data and indeed strengthens our conclusion of little stratospheric smoke in 2023 as shown in the addition of Fig S1, a time series of MLS CO from 40 to 70 ºN in the lower stratosphere. The comments on individual profiles are due to the lower S/N of MLS compared to ACE, but it is true that large perturbations are still satisfactorily measured. We have added this point to lines 87-88:**

"carbon monoxide data from MLS in the lower stratosphere is still useful as complementary data since strong signals from large perturbations would be clearly detected."

Line 132; Figure 1: I'm perplexed by the high August tropopause and the explanation. The twenty August ACE occultations are distributed evenly, longitudinally. For an average tropopause to be 16 km, some individual ones would have to be even higher to compensate for those distant from the Asian Monsoon sector (which represents at most about ¼ of the zonal belt even at its widest, south of 40N). I examined the August 2023 ACE occultations north of 40N and the temperature data. Eyeballing the tropopause from the temperature profile, I estimated an average ~14 km. I also used NCEP Reanalysis data, interpolated in space and time to the ACE occultations, to calculate tropopause height based on the dynamical definition. The average for these occultations was 13.1 km using the threshold: pv=2.5 pvu. If my results are accurate, this disparity calls into question tropopause height calculations here and perhaps elsewhere in the paper. Considering the crucial importance of the tropopause height in this study, I would encourage the authors to check their calculation method and results, and update all the analyses dependent thereon.

**Response: Thank you for highlighting the anomalously high August tropopause height. Firstly, we have added an additional section (Sect. 2.3) to the methods to explain our calculation of the lapse rate tropopause. For the monthly averages as shown in Figure 1, I took an average temperature profile of all occultations in the latitude and time range of interest, from which I calculated the lapse rate tropopause using the described method. This provides a slightly different value (15.89 km) than if I calculate the tropopause from individual occultations and then take the average (14.8 km) which sounds similar to how you estimated the average height to be ~14 km. However, it is important to note that a significant number of individual occultations exhibit tropopause heights above 15 km, so the higher value regardless of calculation method is not an error.**

**I have also calculated a lapse rate tropopause from ERA5 temperature profiles at the location and times of ACE measurements in August, which results in a tropopause height of 14.6 km. This ERA5 value is close to the ACE values, and the fact that the reanalysis product is available on a different spatial grid and temporal resolution (i.e. every hour) compared to ACE occultations can also partially explain the difference. In any case, we show that the August tropopause for the available ACE data in that month is reasonably around 4 km higher than the other monthly tropopause heights. This occurs since there are much fewer measurements in the latitude range of interest, and they are all located between 40 and 45 ºN as opposed to more evenly distributed across the entire latitude range of 40 to 70 ºN as in other months.**

**The tropopause is higher closer to the equator, so this anomaly is not entirely unexpected. This feature of ACE data in August has also been referenced in previous literature (e.g. Doeringer, D., Eldering, A., Boone, C. D., González Abad, G., Bernath, P. F.: Observation of sulfate aerosols and SO2 from the Sarychev volcanic eruption using data from the Atmospheric Chemistry Experiment (ACE), J. Geophys. Res-Atmos., 117, https://doi.org/10.1029/2011JD016556, 2012). Table S1 is included to make clear that the sampling is limited in August, and this point has been highlighted more clearly in the revised paper.**

**We have added these points to lines 143-149:**

"ACE/SCI-SAT1 takes a very limited number of measurements over our latitude range of interest during August as seen in Table S1; only 20 measurements were taken in August 2023, and all on the first couple of days of the month and between 40 and 45º N. This may explain the unusually high tropopause altitude for this month as tropopause altitudes are higher closer to the equator and previous studies with ACE in August also report a higher tropopause height (Doeringer et al., 2012). Additionally, a low number of measurements are likely more susceptible to transient meteorological influences such as the Asian monsoon (Basha et al., 2020). ERA5 reanalysis data collocated at ACE measurement points and times for August yield a similarly high average tropopause height of 14.6 km (Herschbach et al., 2023)."

**For the purpose of monthly averages, showing average profiles and temperatures as in Figure 1 gives a general context of VMR vs. altitude. The tropopause relative analysis in the later parts of the paper handle each occultation and tropopause calculation individually so these methodological differences then become irrelevant (see below).**

Line 132, Figure 1: The August panel is also puzzling in that the 2017 and ACE-lifetime statistical HCN are larger at 16 km w.r.t. the other months. Even the sigma width is larger. Are the August data from other years all subject to a

similar tropopause bulge? Is the entire ACE August 40-70N record that much different than the adjacent months? Because the August pattern in Figure 1 is so strikingly different, it needs to be addressed exhaustively or removed from the analysis.

**Response: The August data record is anomalous given the small number of data points that are only being measured between 40 and 45 ºN, as opposed to larger amounts of data that are more evenly distributed across the entire latitude range of 40 to 70 ºN as in other months. Given this lack of data, we acknowledge that ACE-FTS is limited in providing us with enough data to analyze possible stratospheric smoke in August. Complementary OMPS (and MLS) data help to fill in these gaps. The text has been edited in lines 156-157 to make this clearer:**

"This limited coverage and amount of August data from ACE is supplemented by the higher coverage of OMPS as shown in the following section and MLS in the supplemental information."

Line 138: This is an argument for using a tropopause-relative reference frame. Given the physical reality of the 2023 plume heights congregating near the tropopause, it is recommended that a tropopause-relative analysis be performed as a replacement or complimentary construct.

**Response: We agree that a tropopause-relative analysis is a necessary part of this work, which is why we perform this analysis and show the data in Figure S3 as well as a tropopause-relative analysis for individual occultations (Fig. 3, Fig. S5). However, a non-negligible amount of ACE occultations do not measure low enough into the troposphere to allow for a tropopause-relative VMR profile. In other words, the data stops above the tropopause. In these cases, the tropopause height is not confidently known and reanalysis would only offer an estimate that is not necessarily accurate enough to make claims about stratospheric smoke entry. Given these measurements, less data is captured by focusing only on a tropopause-relative framework. By calculating VMR averages at every altitude and keeping Figure 1 in absolute altitude, we are able to include the maximum amount of data to compare 2023 profiles with previous years. We then also perform a tropopause relative analysis for HCN and aerosol extinction to further identify individual occultation measurements of smoke in the stratosphere.**

**The following text has been added (lines 130-133):**

"We first investigate average monthly HCN volume mixing ratio (VMR) profiles. Some individual occultations do not extend into the troposphere, so monthly averages in absolute altitude capture the maximum amount of data to compare 2023 with the rest of the ACE-FTS measurement period (2004 to 2022). A tropopause-relative framework is used in Sect. 3.2 to detect individual measurements of stratospheric smoke."

Line 138: A possible complicating factor comes to mind while contemplating this analysis. The phenomenon of the double tropopause [e.g. Homeyer et al. (2014; https://doi.org/10.1002/2014JD021485)] could well be a factor in the tropopause-height calculation and assessing 2023 plume tropopause-relative height. It might be worth noting this and expressly stating if/how double tropopause situations are handled in the calculation of tropopause height.

**Response: Thank you for pointing this out and its relevance to this work. Indeed in some temperature profiles we see evidence of a double tropopause. In these cases, our tropopause calculation provides the altitude of the lower tropopause. This point has been clarified in the revised text (lines 115-116):**

"For temperature profiles that exhibit a double tropopause, this method provides the lower tropopause altitude (Peevey et al., 2012; Homeyer et al., 2014)"

Figure 3: These statistics show that the authors have done a trop-relative analysis of all ACE data. So, it should be considered elsewhere.

**Response: This was addressed two comments prior.**

Figure 4: It would be helpful to the broad readership to add a plume-free spectrum for comparison, akin to Figure S4 in Boone et al. (2020).

**Response: Thanks for this suggestion, we have added additional spectra at different tangent heights to Figure 4, which serves two purposes: 1) gives an example of a plume-free spectrum (at higher altitudes), 2) serves as additional evidence that smoke is limited to the lower stratosphere.**

Line 163, "saturated": Saturated with what, or because of what? This needs to be fleshed out because 3 of three distinct smoke plumes shown in Boone et al. (2020) all had a prominent C=O stretch feature. So, the disparity of the present example w.r.t. these published examples raises the question of why it is different. IR spectroscopy of smoke as compared to other particulate compositions is not yet a mature and intuitive topic for the general science audience. Help is required for the benefit of those not attuned to the specifics of this niche ACE-FTS data application.

**Response: Thank you for highlighting the need to provide a more specific description of this saturation effect. We have added this text in lines 182 -185:**

"The C=O stretch, which is also associated with oxygenated smoke particles and identified using a feature around 1750 $cm^{-1}$, cannot be seen because that region is saturated at lower altitudes by strong absorption of water vapor in the same spectral region"

3.2 Occultation ss107570: This section is where the authors state that only one occultation (ss107570) had a demonstrable smoke enhancement above the tropopause. I have examined all the 2023 ACE profiles north of 40N, in concert with tropopause heights, and found several that are as convincing as ss107570. E.g. ss107346 and sr108266. Given the above-mentioned concern with the tropopause-height calculation, I encourage the authors to re-examine the occultation record for other examples of extinction enhancements clearly above the tropopause to find other candidates for stratospheric smoke. This would have two crucial benefits, 1. Additional IR spectra to exploit, and 2. Other candidates for matching the ACE observation to a potential pyroCb. Another suggestion to the authors is to extend their analysis to October ACE data, considering the fact that there were two Canada pyroCb events late in September and the likelihood of ACE stratospheric smoke observations post September.

**Response: We have extended our analysis to include other months, which indeed allowed for identification of more occultations that exhibited elevated aerosol extinction and HCN as noted by the reviewer. This has been added to the text in Figure 3 and Figure S5. We have also implemented a more systematic method for identifying possible smoke by highlighting an orbit number as a "hit" if it exceeds both an HCN VMR and aerosol extinction threshold above the tropopause. Some of these "hits" were determined to be sulfate aerosols if they did not show features of oxygenated organics upon inspection of the infrared spectra, but others do show evidence of oxygenated organics such as in Figure 4.**

Section 3.3: Either this section or section 2.2 need to contain a detailed description of the OMPS analysis method. Much is unknown. How are clouds handled? How is the vertical data ensemble between 11-16 km treated and reduced to single extinction ratio values? How is the molecular extinction calculated? What is the publication background on this method, if any?

**Response: We have added a more detailed description of the OMPS data processing and references with more information in lines 98-102:**

"A cloud detection algorithm detects the highest cloud altitude and flags all aerosol extinction measurements below the peak cloud level, allowing for a cloud-filtered data product (Chen, 2016). The retrieved aerosol-to-molecular extinction ratio analyzed in this work is analogous to an aerosol mixing ratio, see Loughman et al., 2018 and Taha et al., 2022 for a detailed account of the retrieval algorithm and previous data applications"

**We have also specified that we took an average (lines 203-204) in the altitude range of interest (either 10.5-11.5 km or 11.5-16.5 km) for Figures 5, 7, and 8:**

"Aerosol extinction data from OMPS LP also indicate a minor increase in aerosol burden in the lower stratosphere averaged between 11.5 and 16.5 km"

Figure 5: The color range here seems to indicate an average extinction ratio <<1.0. This plot does not conform to Figure 8 (top). The general average between 40-80N is <<1.0 in Figure 5 while Figure 8 shows a baseline near 1.0. Something appears to be amiss.

**Response: Thank you very much for pointing this out. The contour plots were showing the data in log scale, while the color bar was not. Changed plots in decimal scale have been added to the text so there is now consistency between the contour plots and the averaged time series.**

Section 3.4: Suggestion: Move this section to 3.3 to connect it directly to the individual ACE profile presented in 3.2.

**Response: We have added our statements on the trajectories directly after showing the ACE occultations, thank you for the suggestion.**

Figure 6: I don't understand the rationale for launching 3-day trajectories within a 3.5-day span of launch times. How does this "compensate for limited ACE-FTS coverage?" The ACE occultation location and time are known precisely. The back trajectory from ACE time does not nearly extend long enough to implicate any land/fire surface area. Only the trajectories launched ~3 days before ACE pass over a suspect landscape. If one were to launch forward trajectories from these times to ACE time, it is not certain where they'd end up and if/how they would apply to the ACE occultation. Hence, as presented, this analysis is unconvincing. It is suggested that the authors restrict launch times to ACE time and run them back long enough to encounter locations/times of reported pyroCbs.

Figure S4: Ditto the concerns expressed above. In this case there is some wiggle room for initiation time, but only on the order of a couple hours. As it stands, the trajectory launched at pyroCb time ends up somewhat close to the ACE location but 3 days before the ACE occultation.

**Response: We have changed the trajectories so that they are initialized at ACE measurement locations and times, and run for 315 hours (the maximum on NOAA HYSPLIT) to see if they intercept any reported pyroCbs. The closest hit is still tens of kilometers away and a few hours off, but the variability in meteorological data and subtle changes in launch time and altitude suggest these fires likely influence composition as it is measured by ACE. These trajectories have been added to the text in Fig. S6.**

**ACE's limited coverage is an issue for direct detection of pyroCbs as they enter the stratosphere. However, the chemical signatures of the smoke are still detectable as the air disperses in the lower stratosphere as seen in the ACE measurements. This point has been added to lines 199-201:**

"Despite the lack of direct detection, chemical signatures of smoke after dispersion in the lower stratosphere are clearly measured and show the limited vertical range of wildfire influence in the stratosphere: within 2 km above the tropopause."

Line 218, "The maximum VMR and aerosol extinction values are similar between the two events, and Fig. S5 also shows that the aerosol loading near the tropopause is similar between the two years.": This sentence is confusing. Are the data in Fig. 7 taken as representative of PNE and 2023, such that the comparative gas and aerosol data maxima reflect a similarity between the PNE and 2023 "events"? Moreover, the PNE profile's CO max is not similar to the 30 July 2023 CO max. The extinction max for the two are similar, but the value for 2023 is in the troposphere. Substantial elaboration and clarification are called for. This is especially important given that the authors relate their synthesis of Figure 7 to the season-long graphic in Figure S5.

**Response: We have changed this section to show a more general comparison between 2017 and 2023, seen in both the OMPS LP and ACE data. It is clear using both gaseous and aerosol products that the PNE reached much higher levels in the stratosphere than any of the Canadian sector wildfire events in 2023. We have removed the statements on maximum levels and added the following text to lines 223-225:**

"Figure 6 shows a substantial difference in smoke altitude, with the 2017 PNE leading to chemical signatures of biomass burning products measured above 20 km compared to the much lower vertical extent of the 2023 fires."

Line 219-221, "However, the stratospheric impact of the PNE is visibly larger since the entire plume is measured well above the tropopause whereas only part of the Yukon plume is clearly in the stratosphere.": Again, the authors seem to be taking this singular PNE profile as somehow representative of PNE in general, and in terms of injection height specifically. The PNE pyroCbs did not inject smoke to the tropopause+9 km.

**Response: We have changed the wording to emphasize the difference in overall impact and vertical extent of the fires from these two years (same as the text above) as opposed to a comparison of injection heights, which we recognize cannot be directly inferred from ACE measurements. Thank you for this suggestion.**

Figure 8: The pre-PNE color shows extinction ratio <= 0.0 according to the color bar, but the line on the upper panel indicates values exceeding 0.4. Is there a discrepancy here?

**Response: Similar to previous OMPS plot, the data was previously in log scale. It has now been appropriately adjusted.**

Figure 8: The difference in extinction ratio between 2017 and 2023 is striking, from the beginning of May onward. The loading in 2023 is 2 to 3 times larger than 2017. What explains this difference? The earliest Canada pyroCbs were on 4-5 May, so the large 2023 values from onset are difficult to understand. What was the extinction ratio pattern prior to May? Do the authors have an explanation for this apparent puzzle?

**Response: We have changed Figure 5 to showcase a longer time period in 2023, which highlights this phenomenon of larger extinction ratios before the onset of the 2023 Canadian wildfires. This emphasizes another source for aerosols in the Northern Hemisphere midlatitudes from an earlier time period. The attribution of these previous aerosol sources is outside of the scope of this paper, but some studies (Taha et al., 2022) suggest recent volcanic eruptions to be an important source. This can also be seen in Figure S7, where there is a large aerosol burden in January 2023 around 18 km that cannot be attributed to Northern Hemisphere wildfires. This has been added to the text in lines 239-244:**

"It is clear that the active pyroCb activity in the 2023 wildfire season did influence upper tropospheric composition given the increase in aerosol extinction starting in May, which coincides with the start of the burning season. This is in stark contrast to the stratospheric aerosol loading from Figure 5, which is largest at the beginning of the year. Thus, the stratospheric impact of the 2017 PNE is visibly more significant since smoke following the event is measured well above the tropopause, whereas smoke from the 2023 fires remains around the tropopause throughout the entire wildfire season."

Figure 8: The 2017 extinction ratio jumps to values ~1.5 after PNE in the lower panel. But the color-scaled top panel manifests no such value, even at its maximum value. Something is apparently amiss.

**Response: As addressed in previous comments, the data was previously in log scale. It has now been appropriately adjusted.**

Figure S1: If this is to remain in the paper, as a validation of ACE CO, more information is called for. Was there an attempt to match MLS profiles with the ACE profiles? Were all the MLS data, day and night, included in the averaging? If essentially all the MLS data 40-70N are used, it probably doesn't qualify this as an ACE validation.

**Response: The purpose of this figure, now Fig. S2, is not to validate ACE CO but rather to show that despite differences in data coverage, a similar monthly average vertical profile for both ACE and MLS are retrieved in the 40-70 ºN latitude band. We have clarified this purpose in the text in lines 89-90:**

"comparison of ACE and MLS profiles in our latitude range of interest supports our conclusions from ACE data despite more limited coverage (Fig. S2)"

Figure S2: Recognizing the authors' uncertainty about enhanced HCN w/o extinction, certain data points call out for an explanation. By that I mean the gray enhancements, high above the tropopause, exceeding the red-dot enhancements in May, July, August, and September. As such, the reader might wonder about how robust the red dots are or what to make of these gray HCN enhancements. Please explicitly deal with these perplexing data points.

**Response: We have added an additional figure in the supplemental information (Fig. S4) that provides an example of how some occultations exhibit enhancements of HCN above the tropopause without showing enhanced stratospheric aerosol extinction. Discussion of this feature (enhanced HCN without aerosol extinction) has also been added to the supplemental information.**

**The original supplemental figure has also been changed after the calculation of tropopause heights was iterated upon as previously described and detailed in Section 2.3.**

Figure S5. The top two panels are defined as extinction ratios for a single, thin layer between 10.5 and 11.5 km. But the bottom panel is defined as relating to 11.5-16.5 km (like the plots in the main body). Is this the authors' intention? It seems odd.

Figure S5 (bottom panel): The 2023 time series shows a pronounced mound of extinction ratio from mid-July to August. This is not apparent in Figure 5, which is this panel's mate, according to the figure caption. It looks much more like the 2023 panel above. Regardless of which panels are matches, the numeric values in the time series do not conform to the color-scaled plots in either figure. Something is apparently amiss.

How is this panel's construction for 2017 different than Figure 8's top panel? Both are described identically in their captions but the time series lines are different.

**Response: The caption for Fig. S5 (now Fig. 8) had an error, it should have said 10.5-11.5 km instead of 11.5 to 16.5 km, thank you for pointing it out. With the data no longer in log scale, the mound in the time series is now more easily seen in the enhanced aerosol in mid-July to August.**

Figure S5: Logically, this figure—once corrected--belongs in the main paper. It is central to the authors' thesis.

**Response: Thank you for this suggestion, we agree and have moved this figure to the main text (Fig. 8).**

**The minor comments from RC1 were also all addressed and are greatly appreciated, as they provided many opportunities for strengthening the text.**

Referee 2 (RC2):
This study uses satellite remote sensing datasets to examine the vertical extents of wildfire smoke, along with chemical signatures of biomass burning, during the record fire season of 2023 in Canada. This topic has high relevance for a

broad community interested in the role of large wildfires in a warming climate system. However, several issues must be addressed prior to publication in ACP, which amount to a major revision.

While I understand the focus on stratospheric impacts (or the lack thereof), it would be nice to have a bit more focus on the potential implications of having so much smoke in the upper-troposphere. Can we say anything about what this might mean for future climate scenarios?

**Response: Thank you for this comment; we also agree that it is a very interesting and important question to investigate the implications of a large smoke burden in the upper troposphere. Modeling studies aimed at answering this question are outside of the direct scope of this work, but it is certainly a necessary direction for the field. Some examples of previous work that is relevant to this question are now referenced in lines 262-263 and include:**

**Christian, K., Wang, J., Ge, C., Peterson, D., Hyer, E., Yorks, J., and McGill, M.: Radiative Forcing and Stratospheric Warming of Pyrocumulonimbus Smoke Aerosols: First Modeling Results With Multisensor (EPIC, CALIPSO, and CATS) Views from Space, Geophys. Res. Lett., 46, 10061-10071, https://doi.org/10.1029/2019GL082360, 2019.**

**Kochanski, A. K., Mallia, D. V., Fearon, M. G., Mandel, J., Souri, A. H., and Brown, T.: Modeling Wildfire Smoke Feedback Mechanisms Using a Coupled Fire-Atmosphere Model With a Radiatively Active Aerosol Scheme, J. Geophys. Res-Atmos., 124, 9099-9116, https://doi.org/10.1029/2019JD030558, 2019.**

**Li, Y., Dykema, J., Deshler, T., and Keutsch, F.: Composition Dependence of Stratospheric Aerosol Shortwave Radiative Forcing in Northern Midlatitudes, Geophys. Res. Lett., 48, e2021GL094427, https://doi. org/10.1029/2021GL094427, 2021.**

More information is needed in the methods section on how tropopause altitudes were calculated. This should be a stand-alone section. How do the methods used here to obtain tropopause altitude compare with previous studies that used other methods and data sources. There is currently only one sentence on using temperature profiles from ACE-FTS to derive tropopause altitude. How are these profiles derived? How does the accuracy compare with reanalysis or radiosonde temperature profiles? Have any previous studies used ACE-FTS to obtain tropopause altitudes?

**Response: We have added Section 2.3 in the methods to more thoroughly describe how we calculated tropopause height from the WMO lapse rate definition**

**To reiterate the main points here, validation of ACE temperature was published by Sica et al., 2008 (https://doi.org/10.5194/acp-8-35-2008). Use of these temperature profiles to determine tropopause altitudes has been discussed in previous literature (e.g. Sioris et al., 2010; Tereszchuk et al., 2013). In our study, we calculated and interpolated the lapse rate from the ACE temperature profile, and then found the lowest altitude at which the lapse rate decreased below $-2$ K km$^{-1}$ to be the tropopause altitude.**

**Sioris, C. E., Boone, C. D., Bernath, P. F., Zou, J., McElroy, C. T., McLinden, C. A.: Atmospheric Chemistry Experiment (ACE) observations of aerosol in the upper troposphere and lower stratosphere from the Kasatochi volcanic eruption, J. Geophys. Res-Atmos., 115, D2, https://doi.org/10.1029/2009JD013469, 2010.**

**Tereszchuk, K. A., Moore, D. P., Harrison, J. J., Boone, C. D., Park, M., Remedios, J. J., Randel, W. J., and Bernath, P. F.: Observations of peroxyacetyl nitrate (PAN) in the upper troposphere by the Atmospheric Chemistry Experiment Fourier Transform Spectrometer (ACE-FTS), Atmos. Chem. Phys., 13, 5601–5613, https://doi.org/10.5194/acp-13-5601-2013 , 2013.**

Some of the figures in this paper present results with tropopause relative altitude while others use absolute altitude. It might be best to use one of these methods in all figures for consistency. In this case, tropopause relative altitude seems like the better choice?

**Response: See responses to Review 1. A number of ACE occultations do not measure low enough into the troposphere to allow for a tropopause-relative VMR profile (i.e. the data stops above the tropopause). In these cases, the tropopause height cannot be calculated and reanalysis would only offer an estimate that is not accurate enough to claim stratospheric penetration. Given these measurements, less data would be captured by focusing on a tropopause-relative framework. By calculating VMR averages at every altitude and keeping Figures 1 and 2 in absolute altitude, we are able to include the most amount of data to compare 2023 profiles with previous years.**

**However, we do perform a tropopause relative analysis for HCN and aerosol extinction to identify individual occultation measurements of smoke in the stratosphere, as seen in Figure 3, and Figure S3 and S5. In summary, we believe the absolute altitude provides important context for the entire burning season while tropopause-relative analysis is necessary for investigation of individual measurements to determine stratospheric penetration. This point has been clarified in the text (lines 130-133):**

"Some individual occultations do not extend into the troposphere, so monthly averages in absolute altitude capture the maximum amount of data to compare 2023 with the rest of the ACE-FTS measurement period (2004 to 2022). A tropopause-relative framework is used in Sect. 3.2 to detect individual measurements of stratospheric smoke."

The decision to omit MLS from the analysis in this paper needs more clarification. It seems that MLS and ACE-FTS have their own sets of strengths and weaknesses for a study like this. So, using them in combination would likely have benefits. MLS data have also been used in previous pyroCb plume studies. At minimum, the authors should cite some of this previous work as part of a more robust justification for why MLS is not used.

**Response: See responses to Reviewer 1. The reason we did not initially include MLS analysis in this study was that the lowest level for recommended use, 215 hPa, was not quite low enough to make assertions of stratospheric vs. tropospheric smoke as with ACE–FTS. However, we agree that the strengths of MLS such as higher spatial coverage merit further analysis so we have included analysis of MLS CO data in Figure S1. The following text has been added to lines 85-88:**

"Since we are interested in the vertical profiles of biomass burning products, MLS is not as useful as ACE for our analysis given the lack of reliable data at our pressure range of interest in the lowermost stratosphere and its transition to the tropopause and upper troposphere." However, carbon monoxide data from MLS in the lower stratosphere is still useful as complementary data since strong signals from large perturbations would be clearly detected (Fig. S1)."

Fig. 1: Please provide more discussion on the tropopause altitude discrepancy in August. This relates to my tropopause methods comment above. Perhaps run some comparisons with tropopause altitudes derived from reanalysis data, such as MERRA-2. It may also be helpful to include a range or standard deviation marker for the tropopause, along with the mean.

**Response: See responses to Reviewer 1. Thank you for the suggestion to compare the ACE-derived tropopause to reanalysis. We have added a comparison to an ERA5-derived tropopause, and also referenced previous literature that also found a higher tropopause altitude in August given its lack of data coverage at higher latitudes. The following text has been added to lines 143-149:**

"ACE/SCI-SAT1 takes a very limited number of measurements over our latitude range of interest during August as seen in Table S1; only 20 measurements were taken in August 2023, and all on the first couple of days of the month and between 40 and 45º N. This may explain the unusually high tropopause altitude for this month as tropopause altitudes are higher closer to the equator and previous studies with ACE in August also report a higher tropopause height (Doeringer et al., 2012). Additionally, a low number of measurements are likely more susceptible to transient meteorological influences such as the Asian monsoon (Basha et al., 2020). ERA5 reanalysis data collocated at ACE measurement points and times for August yield a similarly high average tropopause height of 14.6 km (Herschbach et al., 2023)."

It seems like the signal from the 2017 PNE is missing in the September panel (blue dashed curve)? I would expect to see a significant enhancement above the tropopause.

**Response: The influence of the 2017 PNE can be seen in the enhancement of HCN above ~15 km that is consistently larger than the data record average + σ. The signal is somewhat muted by the averaging over 40 to 70 ºN, and also the fact that the stratospheric plume measured in early September had a strong signal outside of this latitude range (38 ºN, Boone et al., 2020).**

Fig 5: This might be more informative as an anomaly plot relative to the OMPS data record…or add a second panel with this anomaly information.

**Response: See responses to Reviewer 1. Thank you for this suggestion. We have changed our OMPS plots (Fig. 5, Fig. 7, Fig. 8) to include months prior to the wildfire season so the level of "background" aerosol at the start of 2023 can be seen. This provides important context as it is evident in the lower stratosphere (11.5 to 16.5 km) that there is substantial aerosol present before the wildfires began to really burn (May onwards). Similarly, by comparing May to October with January to May in the UTLS (10.5 km to 11.5 km) as in Figure 8 we see a clearer enhancement starting in May that can be more confidently attributed to wildfires. The following text has been added to lines 239-244:**

"It is clear that the active pyroCb activity in the 2023 wildfire season did influence upper tropospheric composition given the increase in aerosol extinction starting in May, which coincides with the start of the burning season. This is in stark contrast to the stratospheric aerosol loading from Figure 5, which is largest at the beginning of the year. Thus, the stratospheric impact of the 2017 PNE is visibly more significant since smoke following the event is measured well above the tropopause, whereas smoke from the 2023 fires remains around the tropopause throughout the entire wildfire season."

Fig. 6 and S4: The trajectory analysis can be improved. In Fig. 6, why was it decided to use multiple time intervals and constrain the trajectories to three days? The time of the ACE-FTS profile should be exact…there is no need for other times. This smoke could have originated from pyroCb activity more than three days before the ACE-FTS observations time. I recommend extending the duration of the trajectories to see if other fires in Canada might have contributed. For Fig. S4, the timing of the trajectories is also very questionable, when considering that a pyroCb event usually lasts for only a few hours. You can look at satellite imagery of the event to get a sense of how long it persisted, and thus the duration of the smoke injection window to launch trajectories from.

**Response: See responses to Reviewer 1. We have changed the trajectories to extend back in time 315 hours and they are now initialized at the time and location of the ACE measurements. These trajectories do not directly intercept reported pyroCb times and locations, but given the limited coverage of ACE this is somewhat anticipated. Despite this, we are still able to detect the chemical influence of the wildfire season more broadly, and the consensus across all ACE measurements as well as OMPS and MLS is that any stratospheric impact is minor and limited to below around 12 km. This has been added in lines 196-201:**

"Back trajectories initialized from ACE stratospheric smoke measurements on NOAA HYSPLIT with GDAS 1º meteorological data do not directly intercept any of these reported pyroCbs, but this is not surprising given the limited spatial coverage of ACE. Trajectories initialized from two occultations, ss107346 and ss107570, pass within tens of kilometers and a few hours of reported pyroCbs and are shown in Fig. S6. Despite the lack of direct detection, chemical signatures of smoke after dispersion in the lower stratosphere are clearly measured and show the limited vertical range of wildfire influence in the stratosphere: within 2 km above the tropopause."

Fig. 7 legend: please make it clear which profile is 2023 vs. the 2017 PNE (red vs. black). That terminology is easier to digest than the profile numbers. Put the profile numbers in the caption.

**Response: Thanks for this great suggestion. We have changed what is now Figure 6 to make the comparison more clear.**

Fig. 8 and S5: It took me a long time to digest what's going on in these figures. The values of the mean curves in the time series are near the top of the color bar scale in the shaded plots when smoke is present, which doesn't make sense. Are the curves the maximum value? In Fig. S5, the caption notes that different layers are used for these plots, but in Fig. 8 they use the same layer? Regardless, this analysis is critical to the narrative. It should all show in the main paper…not the supplement…after the errors are fixed.

**Response: Thank you, see responses to Reviewer 1. There was an issue with the scaling of data; the plot was showing the values in log scale which is why they did not visually match the time series averages. This has now been fixed so that there is consistency between the contour plots and time series. We have also fixed the typo in the caption of Fig. S5 and moved it to the main text (now Fig. 8).**
* * *
Referee 3 (RC3):
The paper presents an interesting analysis on the presence of particles in the stratosphere from 2023 summer Canadian wildfires, using some satellite data. Nevertheless, the paper is too short, some works on the same subject need to be considered for an improved discussion, and perhaps others sources of data should be analyzed before concluding that such smoke particles from wildfire are rare in the stratosphere.
The paper need a major revised before it can be considered for publication.

**Response: Thank you for your comments. We would like to reiterate that the focus of this study is on the 2023 Canadian wildfire season, and our findings (as supported by OMPS, ACE, and MLS data) point towards minimal stratospheric impact. This is especially true in comparison to previous events such as the 2017 Canadian Pacific Northwest Event which is used as a comparison event throughout the paper.**

**We are not claiming that smoke particles from wildfires are unable to enter the stratosphere, as there are many publications including those that you provide which indicate otherwise. Many of these publications are centered around two very specific events: the 2019-2020 ANYSO and the 2017 PNE.**

**It is seen in the data that smoke from the 2023 wildfires did not have a large stratospheric impact that perturbed composition or would influence chemistry in a meaningful way. Given the large area burned and the record-breaking number of pyroCbs in the 2023 Canadian wildfire season, it may be surprising that the stratospheric impact was minimal and suggests the rarity of such events. This point is emphasized in lines 265-267:**

"In summary, this work shows that despite an extremely extensive wildfire season with frequent pyroCb activity in Canada, the conditions for sufficiently deep convection were met so rarely and to such a limited extent that no significant stratospheric perturbation took place."

**However, we are aware of past examples of stratospheric smoke from previous wildfire events in western North America, and this is also directly shown in data in Sect. 3.4 where we compare 2017 to 2023 such as in lines 242-244 :**

"Thus, the stratospheric impact of the 2017 PNE is visibly more significant since smoke following the event is measured well above the tropopause, whereas smoke from the 2023 fires remains around the tropopause throughout the entire wildfire season."

The authors speak of vertical extend, but no vertical profile of particles in the stratosphere are presented. So, the title of the paper must be changed, or real aerosol vertical data in the stratosphere, perhaps from different instrumental sources, must be considered.

**Response: We have clarified that OMPS measures aerosol extinction to our method section in line 99:**

*"aerosol extinction is analyzed as an indicator for aerosol abundance"*

**We have also added an additional reference to lines 72-75 which describes the aerosol extinction profile retrievals for ACE:**

*"A pair of filtered imagers also measures atmospheric extinction at two wavelengths: visible (VIS, 527.11 nm) and near-infrared (NIR, 1020.55 nm). The NIR imager is less likely to become saturated in cases of strong aerosol extinction and was thus used in this analysis for detection of aerosol loads (Vanhellemont et al., 2008; Boone et al., 2020)."*

**OMPS LP measures aerosol extinction, and the retrieval for this product and its use to analyze aerosols is detailed in numerous publications that are cited in our paper. ACE Imager extinction measurements were also employed to identify smoke signatures. Of course, wildfires are not the only source of particles in the stratosphere, especially in the recent context of the Hunga Tonga–Hunga Ha'apai and Shiveluch eruptions in the last couple of years, which is why we also focus on chemical data from ACE–FTS to be able to confidently attribute wildfires as the source. This point has been added to lines 126-128.**

*"Additionally, wildfires are not the only source of particles in the stratosphere, especially in the recent context of the Hunga Tonga–Hunga Ha'apai and Shiveluch eruptions in the last couple of years. This is why we focus on chemical data from ACE–FTS in addition to aerosol data to be able to confidently attribute wildfires as the source."*

**We have added in vertical profiles from OMPS to our supplemental information (Fig. S7) to complement the latitude vs. time data we already included in our analysis. Thank you for suggesting this addition because it also clearly shows a distinct aerosol source at a higher altitude (~18 km) from before the wildfire season, and that the perturbation from wildfires starting in May is limited to below ~12 km.**

Line 55 : Some publications are missing, in particular the ones showing that large amount of soot particles can be injected in the stratosphere during wildfire events. Also, others works are available on the vertical extend of smokes in the stratosphere. The authors must rewrite this paragraph, considering the recent results on the stratospheric solid particles and on vortex resulting from wildfires. Here some examples of papers:

- Sellitto, Pasquale & Belhadji, Redha & Cuesta, Juan & Podglajen, Aurélien & Legras, Bernard. (2023). Radiative impacts of the Australian bushfires 2019–2020 – Part 2: Large-scale and in-vortex radiative heating. 10.5194/egusphere-2023-1067.
- Lestrelin, Hugo & Legras, Bernard & Podglajen, Aurélien & Salihoglu, Mikail. (2021). Smoke-charged vortices in the stratosphere generated by wildfires and their behaviour in both hemispheres: comparing Australia 2020 to Canada 2017. Atmospheric Chemistry and Physics. 21. 7113-7134. 10.5194/acp-21-7113-2021.
- Renard, Jean-Baptiste, Berthet, Gwenaël, Levasseur-Regourd, Anny-Chantal , Beresnev, S. ,  Miffre, Alain , Rairoux, Patrick , Vignelles, Damien , Jegou, F.. (2020). Origins and Spatial Distribution of Non-Pure Sulfate Particles (NSPs) in the Stratosphere Detected by the Balloon-Borne Light Optical Aerosols Counter (LOAC). Atmosphere. 11. 10.3390/atmos11101031.

**Response****: Thank you for highlighting these works in relation to the subject of our paper, which we agree provide helpful context on the mechanism of past stratospheric wildfire impacts. Other events are not the focus of this paper, nor do our results challenge those conclusions. Here we focus specifically only on the vertical extent of smoke and chemical species from the 2023 Canadian fires.**

**The following statement has been added to our text in lines 38-40:**

*"Strong evidence has been presented to show that past wildfire events injected significant amounts of smoke above the tropopause, and that smoke-charged vortices may also self-loft (Khaykin et al., 2020; Renard et al., 2020; Lestrelin et al., 2021, Sellitto et al., 2023)."*

Can the authors provide error bars for the concentration in Figure 3 and Figure 4? Without such error bars, it is not possible to assess the reality of the detections.

**Response: Statistical errors from the fitting process for retrieved volume mixing ratios are on the order of $10^{-11}$ ppv, which is orders of magnitude lower than the VMR values measured. Previous publications detailing smoke plumes from ACE–FTS have also only shown the retrieved value for visual clarity because of this.**

Line 170-174 : This conclusion seems to be in contradiction with other works on this subject. The author must improve the bibliography and explain how their results can be compared to previous works and why they could differ.

**Response: Here we are only focusing on the 2023 Canadian wildfires, and our results are the first to our knowledge to report on the lack of substantial stratospheric impact from last year's events. This is not in contradiction to reports of previous stratospheric smoke from past wildfire events which we recognize had significant stratospheric impacts as highlighted both in our previous and updated reference list. The question of why similar impacts were not realized for these events is outside the scope of our study but merits further investigation and is likely related to both land and atmospheric influences.**

Line 180: Figure 5 do not show the vertical profile of aerosol extinction, so how the authors can speak of a "significant decrease in signal past 11.5 km"?

**Response: Since Figure 5 shows the average aerosol extinction between 11.5 and 16.5 km and this is compared to the UTLS (10.5 to 11.5 km) average in Figure 8, we can confidently say that aerosol loading is substantially lower above 11.5 km compared to below. To make this clearer we have added in an additional OMPS figure to the supplemental information (Fig. S7) which shows that the aerosol signal possibly attributed to wildfires is limited to below ~12 km.**